# Has the Digital Economy Reduced Carbon Emissions?: Analysis Based on Panel Data of 278 Cities in China

**DOI:** 10.3390/ijerph191811814

**Published:** 2022-09-19

**Authors:** Zhuoxi Yu, Shan Liu, Zhichuan Zhu

**Affiliations:** 1School of Mathematics and Statistics, Liaoning University, Shenyang 110031, China; 2School of Economics, Liaoning University, Shenyang 110136, China

**Keywords:** carbon emission, digital economy, green energy efficiency, decoupling effect, interaction effect, threshold model

## Abstract

China is undergoing an urbanization process at an unprecedented scale, and low-carbon urban development is of great significance to the completion of the “dual carbon goals”. At the same time, the digital economy has become an important engine for urban development, and its role in environmental improvement has become increasingly prominent. While the digital economy is booming, can it promote the low-carbon development of cities? Based on the panel data of 278 cities in China from 2011 to 2019, this paper discusses the impact of the digital economy on carbon emissions and the long-term development trend between the digital economy and carbon emissions, the impact of differences in the development level of the digital economy on carbon emissions reduction, and the impact of green energy efficiency in the relationship between the digital economy and carbon emissions. The results show that the digital economy has a significant inhibitory effect on carbon emissions, and with the development of the digital economy, more and more cities show an absolute decoupling of the digital economy and carbon emissions and are turning to low-carbon development. The development level of the digital economy has a heterogeneous impact on carbon emissions. With the improvement of the development level of the digital economy, the effect on emission reduction is more significant. As a threshold variable, green energy efficiency affects the relationship between digital economy and carbon emissions. When green energy efficiency is low, the digital economy promotes carbon emissions, and when green energy efficiency is high, the digital economy reduces carbon emissions.

## 1. Introduction

The increase in carbon emissions is one of the important factors of global warming [1], leading to frequent natural disasters, food production reduction, ecological environment imbalance, and other problems, which seriously threaten the sustainable development of human society. In recent years, China’s carbon emissions have continued to increase. World Bank data show that China emitted 9809.2 million tons in 2019. China is one of the countries with high carbon emissions in the world. The acceleration of urbanization has significantly increased carbon emissions. Government infrastructure projects, people’s living consumption, and urban transportation consume a lot of energy. Cities have become one of the main sources of carbon emissions growth, accounting for 70% of carbon emissions [2]. At the same time, the rapid development of the digital economy has widely concerned all sectors of society. The “digital economy” was first proposed by the American scholar Tapscott [3]. In the subsequent development process, OECD, the US Department of Commerce, and other authoritative institutions have expounded on the definition and accounting of the digital economy. With the development of information technology and the continuous improvement of economic and social digitization, the connotation and scope of the “digital economy” are being further expanded. In 2016, the G20 summit of Hangzhou adopted the *G20 Digital Economy Development and Cooperation Initiative*, which defined the digital economy as a series of economic activities that includes using digitized information and knowledge as the key factors of production, modern information networks as the important activity space, and the effective use of information and communication technology (ICT) as an important driver for efficiency-enhancing and economic structural optimization. The report on the construction and development process of Digital China (2019) shows that China’s digital economy has shown a rapid development trend, with an added value of RMB 35.8 trillion and a contribution rate of 67.7% to GDP growth. The digital economy plays an important role in reducing regional differences, coordinating regional development, improving economic resilience, and forming a sustainable development ecosystem. It also plays an irreplaceable role in improving the regional environment [4]. Therefore, can the digital economy help reduce carbon emissions? What are the impacts of different levels of the digital economy on carbon emissions? What is the action mechanism of the digital economy on carbon emission reduction? By clarifying the above three issues, the relationship between the digital economy and carbon emissions is clearer, and corresponding policy recommendations are put forward to provide policy references for the completion of the goal of “carbon peaking and carbon neutrality”.

Since the “carbon peaking and carbon neutrality goal” was put forward, the relationship between the digital economy and carbon emissions has become a key topic of concern among all walks of life, and academia has also carried out extensive research. Scholars’ research shows that China’s carbon emissions have increased year by year and show obvious characteristics of uneven spatial distribution. The carbon emissions in eastern China are significantly higher than those in the west [5,6]. China is rich in coal resources and short of natural gas and oil [7]. Therefore, coal has long dominated China’s energy structure [8]. According to World Bank data, China’s coal accounted for 87.4% of primary energy consumption in 2019. The massive use of coal has led to serious environmental problems, and carbon emissions continue to increase [9]. The adjustment of the energy structure, the optimization and upgrading of the industrial structure, and technological progress can promote the use of clean energy, thereby reducing the use of coal and ultimately reducing carbon emissions. Therefore, energy structure adjustment [10,11], industrial structure optimization and upgrading [12,13], and technological progress [14] are effective means to reduce carbon emissions. However, scholars have different research conclusions on the relationship between the digital economy and carbon emissions. Most scholars believe that the digital economy has promoted the development of e-commerce industry and internet industry, squeezed out emission-intensive industries, optimized the urban industrial structure [15,16], improved the environmental situation, and provided impetus for emission reduction. Wang et al. [17] find that the development of the digital economy is conducive to reducing carbon emissions. Zhang et al. [18] believe that the digital economy plays a significant role in carbon emission reduction based on China’s urban data. However, some scholars also believe that the digital economy has a heterogeneous impact on carbon emissions. For example, the research of Xu et al. shows that the digital economy has different impacts on cities in the eastern, central, and western regions of China, significantly inhibiting the eastern region, while promoting the central region [16]. However, some scholars [19,20] believe that, as the core foundation of the digital economy, the development of digital technology will lead to a large amount of power consumption and energy consumption, thereby increasing carbon emissions.

As for the influencing factors of the digital economy, the literature covers a wide range, such as digital infrastructure [21], digital application [21], education [22], science and technology [23], etc. From the existing literature, there are great differences in the development levels of infrastructure, education, technology, and so on among Chinese cities, so there are great differences in the development of the digital economy. The digital development level of the eastern coastal cities is significantly higher than that of the central and western regions, and the polarization characteristics of the eastern and central regions are particularly obvious [24,25]. When digital technology is at different levels of development, it has a different impact on the intelligent and networked development of society and then affects the upgrading of industrial structure and changes the economic paradigm [26]. Under the new economic paradigm, emerging industries have gradually become the pillar industries and leading industries of regional development. Characterized by high technology content and intensive resources, emerging industries have greatly reduced carbon emissions, and this has been confirmed by scholars’ research. The research of Wang et al. [27] shows that the driving effect of the development level of the digital economy on low-carbon development has regional heterogeneity; Li et al. [28] finds that the digital economy has a significant heterogeneous impact on carbon emissions. Different from the eastern and central regions in China, the digital economy in the western region promotes carbon emissions.

Adhering to the idea of low-carbon green development, scholars have included environmental pollution in the analysis of energy efficiency. This input–output efficiency considering the emission of environmental pollutants is defined as green energy efficiency [23]. Green energy efficiency comprehensively considers environmental and resource factors and promotes urban low-carbon development [29]. The digital economy represented by digitalization has affected green energy efficiency [30], thus promoting the transformation of the industrial structure to energy conservation and low carbon. Specifically, digital technology has greatly improved the basic production conditions such as equipment and technology, optimized the factors other than energy input, and promoted the development of green energy efficiency. Areas with better development of the digital economy tend to have developed digital technology, higher green energy efficiency, and higher utilization of clean energy, and have ultimately reduced carbon emissions.

Although scholars have made rich achievements in exploring the relationship between the digital economy and carbon emissions, and have relatively unified opinions on the factors affecting the development of the digital economy, there are still some questions to be answered: the role of the digital economy in reducing carbon emissions needs to be verified; it is necessary to verify the impact of different levels of digital economic development on carbon emissions; and the role of green energy efficiency in the relationship between the digital economy and carbon emissions needs to be clarified. Based on the panel data of 278 cities in China from 2011 to 2019, this paper makes an in-depth study of the relationship between the digital economy and carbon emissions and analyzes the role of green energy efficiency in it.

The rest of this paper proceeds as follows. Section 2 performs the theoretical analysis and research hypotheses; Section 3 introduces the models and related data used in the research methods; Section 4 contains the analysis and discussion of the research results; Section 5 carries out the robustness test; and Section 6 concludes and presents the implications.

## 2. Theoretical Analysis and Research Hypotheses

Tapscott [3] believes that in a modern economy information exists in a digital form. Thanks to digital technology, the digital economy can reduce carbon emissions in many ways. First of all, the digital economy is widely used in life services, industrial development, transportation and so on, penetrating all aspects of production and life. Regional and time constraints are important factors affecting the allocation of resources and factors [31] and are important reasons for the waste of resources and energy. The internet, cloud computing, and big data are the key technologies for the development of the digital economy [32]. These key factors break through the boundaries of region and time, accelerate the flow of factors, ease the constraints of region and time on the allocation of resources and factors [33], reduce the waste of resources and energy, and thus reduce carbon emissions. Second, with the rapid development of the digital economy, the construction of new infrastructure around the country continues to improve. Enterprises, universities, scientific research institutes, governments, and other application subjects can enjoy the convenience brought by the development of new infrastructure. Each application subject applies digital technology to their own development, digitally improves the process, and finally realizes emission reduction. For enterprises, the construction of digital platforms and digital assembly lines can enable industrial processes to achieve self-control and optimization, as well as enterprise resources to achieve coordinated and orderly development, thereby reducing carbon emissions. Universities and scientific research institutes are the inexhaustible driving force of digital innovation [34]. The application of digital technology reduces the cost of knowledge acquisition, facilitates knowledge management, breaks down barriers to information flow, and optimizes the management efficiency of R&D teams, thereby improving digital innovation [35]. Digital innovation achievements are applied to the field of digital technology, breaking through key technical problems, further accelerating the development of the digital economy, and strengthening emission reduction.

The decoupling model is an important method to measure whether there is a synchronous relationship between economic growth and material consumption. It is widely used in the research on the decoupling relationship between economic growth and carbon emissions [36]. Scholar’s research shows that the synchronous relationship between Chinese economic development and carbon emissions is weakened, which is manifested as a decoupling effect [37,38]. From the perspective of the long-term development trend, the development level of China’s digital economy continues to improve, while carbon emissions are gradually reduced. There may not be a synchronous relationship between the digital economy and the long-term trend of carbon emissions. Based on the above analysis, hypothesis 1 is proposed.

**Hypothesis** **1** **(H1)**.*The digital economy significantly reduces carbon emissions and, with the development of the digital economy, the digital economy and carbon emissions show a decoupling effect*.

When the digital economy is in its infancy, local governments increase the construction of new infrastructure, including information infrastructure, innovation infrastructure, and integration infrastructure represented by information centers, big data centers, 5G stations. New infrastructure construction has increased energy consumption, which leads to an increase in carbon emissions to a certain extent [39,40]. When the digital economy develops to a higher level, the role of digital application and digital innovation in industrial development is more obvious, and the industrial structure with low energy consumption and low emissions is further developed to promote the development of a low-carbon economy and reduce carbon emissions [41]. Therefore, carbon emissions may be increased when the development level of digital economy is low, and carbon emissions may be reduced when the development level of the digital economy is high. In addition, the urban grade and spatial distribution reflect the development level of the digital economy to a certain extent, and have different effects on regional carbon emissions. Based on the above analysis, hypothesis 2 is proposed.

**Hypothesis** **2** **(H2)**.*The difference in the development level of the digital economy has a heterogeneous impact on carbon emissions*.

China has a vast territory and a large number of cities. When formulating policies to improve green energy efficiency, local governments are bound to consider regional development conditions, resulting in different policies. Coupled with the distortion and friction of energy prices, green energy efficiency varies significantly across regions [42]. Due to differences in green energy efficiency, the impact of the digital economy on carbon emissions may also be different. On the one hand, regions with low green-energy efficiency are mostly concentrated in central and western China [43], where there are many zombie enterprises or resource-intensive enterprises, which makes it difficult to achieve emission reduction through the digital economy [23]; on the other hand, low green-energy efficiency may affect the development of digitalization and ultimately inhibit the development of a regional digital economy. In other words, regions with higher green-energy efficiency may be more flexible in production, and digital technology is integrated into production as an important factor of production, thereby greatly reducing carbon emissions. Based on the above analysis, hypothesis 3 is proposed.

**Hypothesis** **3** **(H3)**.
*The effect of the digital economy on carbon emission reduction is affected by green energy efficiency. The higher the green energy efficiency, the more obvious the effect of the digital economy on carbon emission reduction.*


## 3. Research Methodology, Variable Selection, and Data Sources

### 3.1. Model Settings

#### 3.1.1. Fixed Effect Model

This paper establishes a panel data fixed-effect model to analyze the impact of the digital economy on carbon emissions. The regression model is shown in Equation (1).
(1)$$CE_{it}=\alpha_{0}+\alpha_{1}EDI_{it}+\rho Z_{it}+\mu_{i}+\varepsilon_{it}$$
where *i* denotes the city and *t* is the year. *CE_it_* and *EDI_it_* represent carbon emission and the comprehensive index of the digital economy development level, respectively. *Z_it_* is a set of control variables, including economic development (GDP), energy structure (Ener), urban construction status (Road), industrial Development (Elec), and financial development level (Fin). *μ_i_* is the individual fixed effect, and *ε_it_* is the random disturbance term.

#### 3.1.2. Decoupling Model

A decoupling index can quantify the relative relationship between economic development and carbon emissions [44]. This paper uses the Tapio model [45] to calculate and classify the relative relationship between the digital economy and carbon emissions, and analyzes the relative relationship between them. The Tapio model is shown in Equation (2).
(2)$$D_{it}^{s}=\frac{\left(CE_{i,t}-CE_{i,t-1}\right)/CE_{i,t-1}}{\left(EDI_{i,t}-EDI_{i,t-1}\right)/EDI_{i,t-1}}$$
where $D_{\mathrm{it}}^{s}$ represents the Tapio decoupling index, and *s* represents the standard decoupling index.

According to the value of the Tapio decoupling index, the decoupling status can be divided into the following eight categories, and the classification is shown in Figure 1.

#### 3.1.3. SBM Model

In this paper, the SBM model considering unexpected output is used to calculate green energy efficiency (*GEE*). Tone [46] improved the traditional data envelopment analysis (DEA) and proposed the SBM model, in which relaxation variables are introduced to accurately measure the efficiency in the presence of unexpected output. The SBM model is a typical nonlinear programming problem. The method proposed by Charnes and Cooper (1962) is used for linear improvement [47], and model (3) is obtained.
(3)$$\begin{array}{c} GEE_{it}=\min(t-\frac{1}{m}\sum_{i=1}^{m}\frac{s_{i}^{-}}{x_{i0}})\\ \left\{ \begin{array}{l} 1=t+\frac{1}{p+q}(\sum_{r=1}^{p}\frac{s_{r}^{g}}{y_{r0}^{g}}+\sum_{r=1}^{q}\frac{s_{r}^{b}}{y_{r0}^{b}})\\ x_{0}t=X\lambda +s^{-}\\ y_{0}^{g}t=Y^{g}\lambda -s^{g}\\ y_{0}^{b}t=Y^{b}\lambda +s^{b}\\ t>0\\ s^{-}\geq 0,s^{g}\geq 0,s^{b}\geq 0,\lambda \geq 0 \end{array}\right. \end{array}$$
where *GEE_it_* are the calculation results of green energy efficiency. *m*, *p*, and *q* are the quantities of input, expected output, and unexpected output, respectively. *s^−^*, *s^g^*, and *s^b^* are the relaxation variables of input, expected output, and unexpected output, respectively. *x*, *y^g^*, and *y^b^* are input, expected output, and unexpected output, respectively, and their element dimensions can be expressed as: *x* ∈ *R^m^*, *y^g^* ∈ *R^p^*, and *y^b^* ∈ *R^q^*, respectively. *X*, *Y^g^*, and *Y^b^* are the matrix composed of *x*, *y^g^*, and *y^b^*, respectively. λ is the weight vector; (*x_0_*, $y_{0}^{g}$, $y_{0}^{b}$) is the decision-making unit.

#### 3.1.4. Interaction Effect Model

Incorporating the interaction term between the digital economy and green energy efficiency into the fixed-effect model can verify whether the role of the digital economy on carbon emissions is affected by green energy efficiency. This paper uses the methods of Peng et al. [48] to build an interaction effect model, and the specific model is shown in Equation (4).
(4)$$CE_{it}=\beta_{0}+\beta_{1}EDI_{it}+\beta_{2}GEE_{it}+\beta_{3}EDI_{it}*GEE_{it}+\varphi Z_{it}+\mu_{i}+\varepsilon_{it}$$
where *EDI_it_*∗*GEE_it_* refers to the interaction term between the digital economy and green energy efficiency. If the regression coefficient of the interaction term is significant, it indicates that the effect of the digital economy on carbon emission reduction will be affected by green energy efficiency.

#### 3.1.5. Threshold Model

Referring to the method of Hansen [49], the indicator function is introduced to construct a threshold model to verify the impact of the digital economy on carbon emissions under different green energy efficiencies. The specific model is shown in Equation (5).
(5)$$CE_{it}=\gamma_{0}+\gamma_{1}EDI_{it}\cdot I(GEE_{it}<\theta_{1})+\gamma_{2}EDI_{it}\cdot I(\theta_{1}\leq GEE_{it}<\theta_{2})+\gamma_{3}EDI_{it}\cdot I(GEE_{it}\geq \theta_{2})+\tau Z_{it}+\mu_{i}+\varepsilon_{it}$$
where *GEE_it_* is the threshold variable. *θ* is the threshold value. *I(·)* is the indicating function; when the expression in brackets is true, the value is 1, otherwise the value is 0. *γ*_1_, *γ*_2_, and *γ*_3_ represent the impact of the digital economy on carbon emissions under different levels of green energy efficiency.

### 3.2. Variable Description

#### 3.2.1. Explained Variable

The explained variable is the carbon emissions (CE). China’s urban carbon emissions include carbon emissions from energy consumption, as well as carbon emissions from power and thermal energy consumption [50,51]. We use the conversion coefficient provided by IPCC 2006 to calculate the carbon emissions of energy consumption [52]. We refer to the methods of Glazer and Kahn [53] to calculate the carbon emissions generated by power consumption. China’s power grid is divided into six regions, and each regional power grid has only one emission coefficient. The benchmark emission coefficient of each regional power grid has been released. Calculate the carbon emissions generated by power consumption in each city by using the benchmark emission coefficient of each regional power grid and urban power consumption. We use urban heating data, thermal efficiency, and the raw coal calorific value coefficient to calculate the amount of raw coal required. According to the carbon emission coefficient per kilogram of raw coal in IPCC 2006, the carbon emissions generated by central heating can be calculated by using the amount of raw coal consumed by thermal energy [50].

#### 3.2.2. Explanatory Variable

The explanatory variable is the comprehensive index of the digital economy development level (EDI). Referring to the relevant literature [21,22,23,54,55], the four dimensions of digital infrastructure, digital development status, digital development vitality, and digital development application are selected to build the index system, and the entropy method is used to measure the comprehensive index of the digital economy development level. The indicator system is shown in Table 1.

The rise of digital economy benefits from the wide application of digital technology [21]. Improving the construction of digital infrastructure is not only the premise of the application of digital technology but also the condition for the rapid development of digital economy. Digital infrastructure indicators include two secondary indicators: mobile phone penetration rate and internet penetration rate, which can directly reflect the development level of regional informatization and the application degree of digital technology in society [23,52]. The digital development status is an indicator that directly reflects the degree of local digital development. R&D expenditure shows the importance that the government and enterprises attach to the development of science and technology. The total amount of telecommunications business shows the degree of integration between the internet and traditional industries, and the number of employees in related industries reflects the current industry capacity. It is a measurement index that reflects the “hard power” of high-tech industries. Digital development vitality is an inexhaustible driving force for the development of the digital economy and an indispensable dimension in the comprehensive index system. The stronger the innovation ability, the greater the informatization development potential. The indicators of digital development vitality are mainly expressed by the proportion of education expenditure, the average number of college students per 100,000 population, and the number of newly established information transmission, software, and information technology service enterprises each year, which can reflect the importance attached to innovation by the local government and scientific research institutes and can also reflect the local innovation output capacity [22]. The digital development application [21] mainly examines the application of digital technology by finance and listed enterprises, including two secondary indicators: the digital inclusive financial index and the digital word frequency of listed companies. The digital inclusive finance index reflects the development level of regional digital finance and green finance. This paper selects the word frequencies of “digital economy”, “cloud computing”, “big data”, “blockchain”, “artificial intelligence”, “AI”, “data mining”, and so on. The number of digital word frequencies of listed companies obtained by the crawler of the annual reports of Listed Companies in China from 2011 to 2019 is an important indicator of the importance of listed companies to the digital economy.

#### 3.2.3. Threshold Variable

In this paper, the total fixed investment, the number of employees at the end of the year, and industrial power consumption are selected as input indicators. GDP is selected as the expected output indicator. Industrial wastewater emissions, industrial sulfur dioxide emissions, and industrial dust emissions are selected as unexpected indicators. The SBM model is used to calculate green energy efficiency (GEE). Green energy efficiency comprehensively considers the input, expected output, and unexpected output of resources and fully considers the utilization of resources and environmental costs. The higher the measured value, the higher the green energy efficiency, and vice versa. In the study of the relationship between the digital economy and carbon emissions, green energy efficiency is taken as a threshold variable to verify the impact of the digital economy on carbon emissions when green energy efficiency is different.

#### 3.2.4. Control Variables

To more accurately analyze the impact of the digital economy on carbon emissions, the following control variables are set: (1) economic development level (GDP), expressed in GDP; (2) energy structure (Ener), expressed by coal consumption per RMB 10,000 of GDP; (3) urban construction status (Road), expressed by per capita road area; (4) industrial development (Elec), expressed by the ratio of industrial power consumption to GDP; and (5) financial development level (Fin), expressed by the ratio of financial deposits to GDP at the end of the year.

### 3.3. Data Source

Based on the relevance, continuity, and authenticity of data, this paper establishes a panel dataset with 278 cities in China from 2011 to 2019 to empirically investigate the impact of the digital economy on carbon emissions and its mechanism. The digital inclusive finance index is from the digital finance research center of Peking University. Other data are from the China Urban Statistical Yearbook, the statistical yearbooks of various cities, and the statistical bulletin of national economic and social development. Some missing data are supplemented by linear interpolation.

Considering the data fluctuation of different dimensions, logarithms are taken for carbon emissions and economic development to solve the heteroscedasticity problem. Before regression analysis of panel data, a unit root test is needed to ensure data stability. LLC, ISP, and Fisher tests show that the data are stable. The average VIF value is 1.66, and there is no significant multicollinearity. Table 2 shows the descriptive statistical results of the main variables.

## 4. Empirical Results and Analysis

### 4.1. Regression Analysis of Benchmark Model

The result of the Hausman test is 254.64 (0.000), which shows that the fixed-effect panel data model is selected for regression modeling. Table 3 lists the regression results of the fixed-effects model of the digital economy on carbon emissions. The results show that the digital economy reduces carbon emissions regardless of whether control variables are added or not and has passed the significance test. The level of economic development and the level of financial development play important roles in promoting carbon emissions. This may be because cities with better economic development vigorously develop the digital economy and increase investment in urban municipal construction and new infrastructure construction, thus increasing the consumption of energy and chemical products, resulting in an increase in carbon emissions. The energy structure and industrial development are closely related to the energy consumption and industrial structure of the city. The unreasonable energy structure and the imbalance of the industrial structure will lead to the production of a large amount of carbon dioxide, which will greatly promote carbon emission.

### 4.2. Analysis of Decoupling Effect

The decoupling index between the digital economy and carbon emissions is calculated through the decoupling model. The range of the coupling index corresponds to the type of urban development one by one. The results are shown in Table 4. The results show that the number of cities that are absolutely decoupled has experienced a change of first decreasing and then increasing. On the whole, carbon emissions have been significantly decoupled from the development level of the digital economy in the past decade. There are more and more cities where, from extensive expansion to low-carbon development, the results of carbon emission reduction are remarkable. The benchmark regression results and decoupling-effect results jointly verify hypothesis 1.

### 4.3. Heterogeneity Analysis

Due to the difference of regional resource endowment and urban development level, there are obvious differences in the development level of the digital economy between cities. Therefore, this paper studies the heterogeneous impact of the digital economy on carbon emissions according to the development degree of cities. According to the spatial distribution, cities are divided into eastern cities, central cities, and western cities (see Appendix A for provincial distribution of cities in each region), and the impact of the digital economy on carbon emissions is discussed respectively. The results are shown in columns 1–3 of Table 5. First, finance divides all cities in China into first tier cities, new first tier cities, second tier cities, third tier cities, fourth tier cities, and fifth tier cities. In this paper, first tier cities, new first tier cities, second tier cities, and third tier cities are defined as more developed cities, and fourth tier and fifth tier cities are defined as less developed cities. The impact of the digital economy on carbon emissions is discussed respectively. The results are shown in columns 4–5 of Table 5.

Columns 1–3 of Table 5 are the fixed-effect-model regression results of the digital economy and carbon emissions in the eastern, central, and western cities of China. From the results, it can be seen that the regression coefficient of the digital economy on carbon emissions in eastern cities is −0.5907, which significantly reduces carbon emissions. The role of the digital economy in central cities is smaller and not significant compared to that in eastern cities. The digital economy in western cities promotes carbon emissions. Columns 4–5 represent the fixed-effect-model regression results of more developed cities and less developed cities, respectively. It can be seen that the digital economy of more developed cities significantly reduces carbon emissions, while the development of the digital economy of less developed cities has a negative effect on carbon emissions, but it is not significant. To verify the impact of the digital economy on carbon emissions in the fourth and fifth tier cities, fifth tier cities are regressed separately. The results are shown in column 6 of Table 5. The results show that the digital economy promotes carbon emissions in the least developed cities in China, which confirms the heterogeneous impact of the development of the digital economy on carbon emissions and verifies hypothesis 2. The reason for the heterogeneity may be that the eastern cities and more developed cities have had a relatively high level of digital development earlier and use clean energy more effectively, so they have a significant negative effect on carbon emissions. In addition, most of the less developed cities are concentrated in the western and northeast regions of China. Most of these regions take firewood and coal as the main heating sources to ensure daily life, which is also an important source of carbon emissions in the less developed regions.

### 4.4. Regression Analysis of Interaction Effect Model and Threshold Model

To verify the role of green energy efficiency in the impact of the digital economy on carbon emissions, the interactive effect model and threshold model are used for verification. The results are shown in Table 6. From the results in column 1 of Table 6, the estimated coefficients of the digital economy and green energy efficiency are −2.1546 and −0.7926, respectively, which have passed the significance level test of 1%, indicating that the digital economy and green energy efficiency have significantly reduced carbon emissions. The coefficient of the interaction term between the digital economy and green energy efficiency is significantly positive, indicating that green energy efficiency strengthens the role of the digital economy in promoting carbon emission reduction, and the role of the digital economy in carbon emission is affected by green energy efficiency.

This paper selects green energy efficiency as the threshold variable and the digital economy as the threshold dependent variable to explore the specific impact of the digital economy on carbon emissions under different levels of green energy efficiency. First, we verified whether there is a threshold effect according to the bootstrap method. The results are shown in Table 7. The results show that the F statistic of the dual threshold model with green energy efficiency as the threshold variable is significant and the triple threshold model is not significant, indicating that there are two thresholds between the digital economy and carbon emissions, which are 0.1872 and 0.2840, respectively.

According to the threshold value obtained, the total sample is divided into three section samples of low efficiency of green energy, medium efficiency of green energy, and high efficiency of green energy. The threshold model is used to test the differential impact of the digital economy on carbon emissions in different sections of green energy efficiency. The results are shown in column 2 of Table 6. The results show that when green energy efficiency is in the low range, the impact of the digital economy on carbon emissions is significantly positive; with the improvement of green energy efficiency, the impact of the digital economy on carbon emissions turns to inhibition. When green energy efficiency is at a medium level, the digital economy reduces carbon emissions, but the emission reduction effect is not significant; When green energy efficiency develops to a high level, the digital economy can significantly inhibit carbon emissions, which verifies hypothesis 3. The main reason for the existence of a threshold effect is that when green energy efficiency is low, the extensive development of cities is more serious, and most enterprises are still in the digital start-up or transformation stage, so green energy efficiency has difficulty playing a role in it. When green energy efficiency develops to a certain level, and the digital economy is integrated into the process of industrial development, the role of green energy efficiency on carbon emissions becomes more and more significant, and ultimately significantly reduces carbon emissions.

## 5. Robustness Test

### 5.1. Replace Econometric Model

To test the effectiveness of the fixed-effect model, this paper uses the GMM method for verification, and the results are shown in column 1 of Table 8. The results of AR (1) and AR (2) show that there is no second-order sequence correlation in the random disturbance term of the first-order difference equation. The Hansen test shows that there is no over-identification problem. The regression coefficient of the GMM method for the digital economy and carbon emissions is estimated to be −0.9936, which has passed the 10% significance test, indicating that the digital economy has significantly reduced carbon emissions. The regression coefficient between the digital economy and carbon emissions measured by the fixed-effect model is −0.9237. The coefficients estimated by the two regression methods are not much different, indicating that the fixed-effect model is effective and the results are robust.

### 5.2. Replace Variable Calculation Method

To evaluate the robustness of the model, this paper replaces the calculation method of the comprehensive index of the development level of the digital economy, adopts the factor analysis method to calculate the digital economy (EDI_fa), and incorporates it into the fixed-effect model. The results are shown in column 2 of Table 8. The results show that the regression coefficient of the digital economy is −0.0918 and has passed the significance test of 5%. The digital economy has significantly reduced carbon emissions, which is consistent with the regression result of the fixed-effect model, and the result is robust.

### 5.3. Replace Explanatory Variable

The “Broadband China” policy is an important measure to fully apply network infrastructure to the development of urban technology to promote the rapid development of the digital economy. This paper uses the policy effect of “Broadband China” to replace the digital economy to evaluate the policy effect and verifies the robustness of the results. The State Council issued the measures for the implementation of the “Broadband China” strategy in 2013, and set up pilot cities in 2014, 2015, and 2016, respectively. Based on the practice of Shi et al. [56], the first batch of pilot cities in 2014 were taken as the experimental group, and the pilot cities in 2015 and 2016 were deleted from the sample to ensure that the results of the DID method and PSM-DID method are the net effect (DID) of the “Broadband China” policy in 2014. The results are shown in columns 3 and 4 of Table 7. In the third and fourth columns, the DID and PSM-DID methods are used to evaluate the policy effect. The results show that the coefficients of the DID and PSM-DID methods for policy effect evaluation are −0.0778 and −0.0765, respectively. Through the 10% significance test, it shows that the “Broadband China” policy has a significant negative impact on carbon emissions, which is consistent with the regression results of the benchmark model, proving the robustness of the results.

### 5.4. Instrumental Variable Method

To effectively alleviate the endogenous problem caused by two-way causality, this paper uses the method of Shen et al. [57] for reference and uses the two-stage least square method for testing. From the perspective of the development process of China’s digital economy, the internet was accessed through telephone lines, and areas with high penetration of fixed-line telephones are likely to have high penetration of the internet. One of the jobs of the post office in the last century was to lay fixed-line telephones. The number of post offices also affects the popularity of fixed-line telephones to a certain extent. The total volume of the post and telecommunications business is a measure of the development of the post and telecommunications, which can integrate the penetration of fixed-line telephones and the distribution of post offices. Carbon emissions are mainly closely related to energy use, technological progress, and economic development, while the total volume of the post and telecommunications business has basically no impact on carbon emissions. Based on the above analysis, the total volume of the post and telecommunications business in 1999 is selected as the instrumental variable of the digital economy.

Drawing on the methods of Nunn and Qian [58], this paper constructs the interaction term between the total per capita post and telecommunications business in 1999 (related to individuals) and the fixed-asset investment in the national information transmission, computer service, and software industry in the previous year (related to time) as an instrumental variable (Instr), and uses two-stage least square estimation to test. The results are shown in Table 9. The results of the KP-LM statistics show that there is no problem of insufficient identification of instrumental variables. CD Wald statistics are significantly greater than the critical value of 16.38 at the 10% level; there is no problem of weak instrumental variables. Therefore, the instrumental variables constructed in this paper are effective. The results in column 2 of Table 9 show that the digital economy has significantly reduced carbon emissions. The regression results considering instrumental variables are consistent with the benchmark regression results, indicating that the results of this paper are still robust after considering endogenous problem.

## 6. Conclusions and Policy Implications

### 6.1. Conclusions

This paper mainly studies the impact of the digital economy on carbon emissions and their long-term development trends, the heterogeneous impact of the difference in the development level of the digital economy on carbon emission reduction, and the role of green energy efficiency in the relationship between the digital economy and carbon emissions. Therefore, based on the panel data of Chinese cities from 2011 to 2019, this paper constructs an index system to measure the comprehensive development index of the digital economy, and uses a fixed-effect model and threshold model to test the impact of the digital economy on carbon emissions and the role of green energy efficiency in multiple dimensions. The main research conclusions are as follows:

First, the development of the digital economy can significantly reduce carbon emissions. After a series of tests, the results are robust. The regression results show that the digital economy has a significant inhibitory effect on carbon emissions, while the levels of economic development, financial development, energy structure, and industrial development have a significant role in promoting carbon emissions. This may be due to the increased investment in new infrastructure construction, which increases carbon emissions, or it may be due to the imperfect industrial structure, which leads to a large amount of carbon emissions. The results of the decoupling effect show that with the development of the digital economy more and more cities have absolute decoupling, from extensive expansion to low-carbon development.

Second, due to the differences in the development level of the digital economy among cities, the impact on carbon emissions is also different. When the development of the digital economy is at a low level, the digital economy increases carbon emissions. When the digital economy develops to a higher level, the digital economy reduces carbon emissions. The results of heterogeneity analysis also show that the development level of the digital economy has different effects on carbon emissions. This may be related to the stage of the development of the digital economy. When the digital economy is in its infancy, a large amount of investment leads to an increase in carbon emissions. With the development of the digital economy, the dividends of the digital economy on carbon emissions gradually emerge and reduce carbon emissions.

Third, the interactive-effect model shows that the effect of the digital economy on carbon emissions is affected by green energy efficiency. When green energy efficiency is in a low sector, the digital economy promotes carbon emissions to a certain extent. However, when green energy efficiency develops to a higher level, the digital economy can significantly inhibit carbon emissions.

### 6.2. Policy Implications

The research conclusion of this paper has a reference role for unswervingly developing the digital economy to reduce carbon emissions.

First, while vigorously developing the digital economy, cities across the country should pay attention to improving the quality of development. China has reached a consensus to accelerate the development of the digital economy from top to bottom, but it does not pay enough attention to the quality of the digital economy development. For the further development of China’s digital economy, the government should combine local advantages, play a guiding and supporting role, create a good business environment, further attract foreign investment, absorb foreign advanced digital technology and ideas, give full play to the spillover effect of knowledge and technology, improve the quality of the labor force, accelerate the application of 5G commercialization, and stimulate enterprises to explore the application scenarios of the digital economy to further release the dividends of the digital economy and lead the digital economy to a higher level.

Second, there is a large gap in the development of the digital economy among cities in China, and the different development levels also have a huge heterogeneous impact on carbon emissions. Therefore, recognizing the current situation of regional development and adapting measures to local conditions are important prerequisites for the development of the digital economy. Build new urban agglomerations around high-level cities with digital economic development to drive the development of the surrounding cities. For low-level cities with digital economy development, although there is a temporary increase in carbon emissions we cannot ignore the huge dividends brought by the development of the digital economy. The development of the digital economy requires leading enterprises to take the lead and expand the digital industrial chain. In particular, high-tech enterprises should be given incentives and loan interest subsidies for key industries and projects.

Third, enterprises should form the concept of green development, integrate digital technology into industrial development, speed up the transformation of achievements, and improve the utilization rate of green energy. Focus on low-carbon-energy technology and green digital technology to improve the effect of technology emission reduction. Support the digital transformation of enterprises and increase the use of clean energy. In the process of promoting the development of the digital economy, we should break the regional and time constraints, take urban agglomeration as the basic unit, improve the collaborative governance capacity of the digital economy, promote regional common development, make the digital economy an important engine of carbon emission reduction, and speed up the realization of the goal of carbon neutrality.

## Figures and Tables

**Figure 1 ijerph-19-11814-f001:**
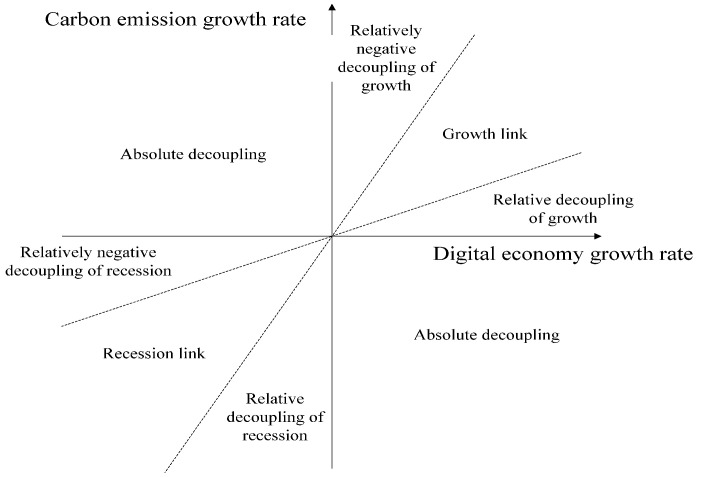
Classification diagram of decoupling state.

**Table 1 ijerph-19-11814-t001:** Comprehensive index of digital economy development level.

Primary Index	Secondary Index	Third Index
Digital economy development level	Digital infrastructure	Mobile phone penetration rate
		Internet penetration rate
	Digital development status	R&D expenditure
		Total amount of telecommunications business
		Number of employees in related industries
	Digital development vitality	Proportion of education expenditure
		Average number of college students per 100,000 population
		Number of newly established information transmission, software, and information technology service enterprises
	Digital development application	Digital inclusive financial index
		Digital word frequency of listed companies

**Table 2 ijerph-19-11814-t002:** Descriptive statistical results of main variables.

Variables	Observations	Mean	Std.Dev.	Minimum	Maximum
CE	2502	6.0708	1.0792	2.4511	10.0372
EDI	2502	0.0678	0.0821	0.0080	0.8774
GDP	2502	16.4444	0.9325	12.7644	19.4112
Ener	2502	0.0933	0.3403	0.0041	14.9024
Road	2502	5.0855	6.4612	0.1812	73.0424
Elec	2502	0.0585	0.1127	0.0004	2.3764
Fin	2502	1.6744	1.3693	0.3711	24.8000

**Table 3 ijerph-19-11814-t003:** Benchmark regression results.

Variables	(1)	(2)
	CE	CE
EDI	−2.1049 *** (−3.29)	−0.9237 * (−1.82)
GDP		0.3559 *** (7.21)
Ener		0.2941 *** (4.93)
Road		0.0067 (1.38)
Elec		0.4755 ** (2.43)
Fin		0.0596 ** (2.50)
Constant	−2.1049 *** (−3.29)	0.0903 (0.11)
City FE	Yes	Yes
Year FE	No	No
R-squared	0.3446	0.6855
Observations	2502	2502

Note: *, **, and *** represent the 10%, 5%, and 1% significance levels, respectively.

**Table 4 ijerph-19-11814-t004:** Results of decoupling effect.

Year	Decoupling Index Range	Form of Expression	Development Type	EDI-CE
2011–2013	(−∞, 0)	Absolute decoupling	Low-carbon development	191
	[0, 0.8)	Relative decoupling of growth	Intensive expansion	30
	[0.8, 1.2)	Growth link	Inefficient expansion	4
	[1.2, +∞)	Relatively negative decoupling of growth	Extensive expansion	53
2014–2016	(−∞, 0)	Absolute decoupling	Low-carbon development	143
	[0, 0.8)	Relative decoupling of growth	Intensive expansion	52
	[0.8, 1.2)	Growth link	Inefficient expansion	14
	[1.2, +∞)	Relatively negative decoupling of growth	Extensive expansion	69
2017–2019	(−∞, 0)	Absolute decoupling	Low-carbon development	208
	[0, 0.8)	Relative decoupling of growth	Intensive expansion	33
	[0.8, 1.2)	Growth link	Inefficient expansion	11
	[1.2, +∞)	Relatively negative decoupling of growth	Extensive expansion	26

**Table 5 ijerph-19-11814-t005:** Heterogeneity impact results.

Variables	(1)	(2)	(3)	(4)	(5)	(6)
	CE	CE	CE	CE	CE	CE
EDI	−0.5907 * (−1.67)	−0.2574 (−0.15)	0.7974 (0.30)	−0.9352 * (−1.80)	−0.0623 (−0.10)	0.8842 (0.54)
GDP	0.5745 *** (12.90)	0.3118 *** (4.63)	0.4662 *** (3.69)	0.5161 *** (9.01)	0.4299 *** (7.94)	0.5097 *** (5.64)
Ener	5.7145 *** (4.57)	0.2956 *** (6.38)	0.2840 *** (11.67)	4.6919 *** (5.56)	3.3923 *** (5.49)	0.2831 *** (19.78)
Road	0.0080 *** (2.75)	−0.0010 (−0.15)	−0.0115 (−1.60)	0.0121 ** (2.60)	0.0174 (1.48)	−0.0174 ** (−2.02)
Elec	0.7627 (1.06)	0.3993 ** (2.30)	0.1352 (1.14)	0.5089 (1.52)	0.5732 ** (1.91)	0.1045 (1.65)
Fin	0.0030 (0.16)	0.0477 * (1.76)	0.1398 ** (2.52)	−0.0422 (−1.55)	−0.0098 (−0.48)	0.1510 *** (4.51)
Constant	−3.5700 *** (−4.31)	0.5082 (0.45)	−2.1078 (−1.02)	−2.4752 ** (−2.44)	−1.5097 (−1.63)	−2.7413 * (−1.84)
City FE	Yes	Yes	Yes	Yes	Yes	Yes
Year FE	No	No	No	No	No	No
R-squared	0.8967	0.3759	0.2514	0.8806	0.9312	0.3691
Observations	990	981	531	1035	1467	747

Note: *, **, and *** represent the 10%, 5%, and 1% significance levels, respectively.

**Table 6 ijerph-19-11814-t006:** Regression results of interaction effect model and threshold model.

Variables	(1)	(2)
	CE	CE
EDI	−2.1546 *** (−3.47)	
GEE	−0.7926 *** (−5.00)	
EDI∗GEE	2.3405 *** (3.05)	
EDI (Th < θ1)		2.3137 *** (3.32)
EDI (θ1 ≤ Th < θ2)		−0.3092 (−0.56)
EDI (Th ≥ θ2)		−1.3798 ** (−2.58)
GDP	0.4384 *** (8.76)	0.4227*** (17.97)
Ener	0.3006 *** (5.03)	0.2980 *** (15.06)
Road	0.0052 (1.15)	0.0061 * (1.91)
Elec	0.3983 ** (2.30)	0.4268 *** (5.57)
Fin	0.0468 ** (2.13)	0.0513 *** (7.75)
Constant	−0.9411 (−1.14)	−0.9880 ** (−2.50)
City FE	Yes	Yes
Year FE	No	No
R-squared	0.7184	0.6621
Observations	2502	2502

Note: *, **, and *** represent the 10%, 5%, and 1% significance levels, respectively.

**Table 7 ijerph-19-11814-t007:** Threshold test results of green energy efficiency.

Threshold Variable	Threshold Test	F-Statistic	*p*-Value	Critical Value		
				1%	5%	10%
GEE	Single threshold	50.58	0.0015	21.9011	26.7847	39.0671
	Double threshold	23.35	0.0535	19.0074	23.6796	26.5915
	Single threshold	50.58	0.0035	23.3312	27.5340	38.3423
	Double threshold	23.35	0.0455	18.4858	23.1141	35.6531
	Triple threshold	10.41	0.5910	23.5394	27.6029	38.6040

Note: *p*-value and critical value are obtained by repeated sampling 2000 times with the bootstrap method.

**Table 8 ijerph-19-11814-t008:** Robustness test results.

Variables	(1)	(2)	(3)	(4)
	CE	CE	CE	CE
EDI	−0.9936 * (1.74)			
EDI_fa		−0.0918 ** (−2.14)		
DID			−0.0778 * (−1.75)	−0.0765 * (−1.68)
GDP	0.9892 *** (16.46)	0.3092 *** (5.54)	0.4224 *** (8.26)	0.4350 *** (8.33)
Ener	0.4014 * (1.82)	0.2944 *** (5.01)	0.1474 *** (3.42)	0.1332 *** (3.34)
Road	0.0410 *** (3.39)	0.0048 (0.93)	0.0130 (1.11)	0.0144 (1.15)
Elec	1.9855 *** (3.67)	0.4485 ** (2.46)	0.4284 ** (2.53)	0.4524 *** (2.65)
Fin	0.0969 *** (2.98)	0.0488 ** (2.03)	0.1481 *** (5.36)	0.1533 *** (5.24)
Constant	−10.6737 *** (−10.60)	0.8418 (0.89)	−1.2316 (−1.43)	−1.4408 (−1.64)
City FE	Yes	Yes	Yes	Yes
Year FE	No	No	No	No
AR(1)	0.000			
AR(2)	0.505			
Hansen	0.781			
R-squared		0.6901	0.6912	0.6835
Observations	2224	2502	1872	1813

Note: *, **, and *** represent the 10%, 5%, and 1% significance levels, respectively.

**Table 9 ijerph-19-11814-t009:** Results of instrumental variable method.

**Variable** **s**	**First-Stage**	**Second-Stage**
	**CE**	**CE**
EDI		−0.9025 *** (−2.88)
Instr	1.02 × 10^−8^ *** (8.55)	
GDP	0.0532 *** (23.21)	0.8545 *** (34.38)
Ener	0.0049 (1.47)	0.2047 (1.52)
Road	0.0011 ** (2.33)	0.0326 *** (8.63)
Elec	−0.0117 (−0.58)	3.3471 *** (6.90)
Fin	0.0141 *** (6.21)	0.0982 *** (3.50)
Constant	−0.8519 *** (−22.47)	−8.3473 *** (−19.82)
City FE	Yes	Yes
Year FE	No	No
KP-LM		8.411 (0.0037)
CD_Wald F		546.141 (16.38)
R-squared	0.6723	0.6700
Observations	2034	2034

Note: **, and *** represent the 5%, and 1% significance levels, respectively.

## Data Availability

The data involved in this study are all from public data.

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
