# Peer review of "Has the Digital Economy Reduced Carbon Emissions?: Analysis Based on Panel Data of 278 Cities in China"

_ijerph, 2022, doi:10.3390/ijerph191811814_

Round 1

Reviewer 1 Report

General comments:

- I do understand how the authors can say that “the digital economy has a significant inhibitory effect on carbon emissions” (in the abstract) while they did not consider indirect carbon emissions due to the digital economy”. In addition, their results show that it depends upon the development stage of digital economy (at the beginning, digital economy is related to increased carbon emissions).

 - Overall, more general theories about the digital economy should be cited, e.g., https://www.sciencedirect.com/science/article/pii/S2210422415300150?casa_token=nd1MgteznRsAAAAA:rP_U0cQig798rzPKuriC2naTn1_euNU1AoApZvMsOSb3LXnAwyylrELTq0uofbkjPQfsPyFaNlNL

https://journals.sagepub.com/doi/full/10.1177/1329878X19844022

1. Introduction

l.37 : Could the authors define what they mean by ‘digital economy’ ? Is that “energy structure adjustment [5-6], industrial structure optimization and upgrading [7-8], technological progress [9] are considered to be effective means to reduce carbon emissions » (l-58-60) ? As described in the methodology section (from line 245), the operationalization used by the authors appears to be broader (educational variables are for instance considered). It is necessary to mention these subdimension in the theoretical introduction.

l.47, and l.107-109: sometimes the authors imply that there is a causal relationship between digital economy and carbon emissions, and sometimes a more bivariate relationship. This needs to be clarified throughout the paper (e.g., “the relationship between digital economy and urban carbon emissions needs further research and verification; the impact of the development level of digital economy on carbon emissions needs to be clarified”).

l.53: Could the authors define what they mean by ‘double carbon goal’? Do they refer to “carbon neutrality and carbon peaking”?

2. Theoretical Analysis and Research Hypotheses

l.125-126: what does “Internet and Internet of things” mean ?

l.150-151: “There is no synchronous relationship between the digital economy and the long-term trend of carbon emissions, and showing a decoupling effect.” The authors should cite some references here (and I think that ‘and’ is not necessary in the last part of the sentence).

167-168: I have two concerns regarding Hypothesis 2 (i.e., “The difference in the development level of digital economy has a heterogeneous impact on carbon emission reduction”). First, prior to describing this second hypothesis, the authors describe this as a fact proven by previous studies (i.e., “The development of the digital economy has a heterogeneous impact on carbon emis- 155 sions [19-20]", l.155.156). If so, why do they propose it as an hypothesis to be tested with their data ? Second, I do not understand what the hypothesis means. Do they authors suggest that the relationship between the digital economy and carbon emissions is not linear ?

3. Research Methodology, Variable Selection, and Data Sources

l.239: "Carbon emissions (CE). Carbon emissions include carbon emissions from direct energy consumption, as well as carbon emissions from electricity and heat consumption" Why did the authors not consider indirect emissions related to the development and implementation of the digital economy ?

l.246: “The comprehensive index of digital economy development level (EDI)” This is not a complete sentence.

l.266-267: “The indicators of digital development vitality are mainly expressed by the proportion of education expenditure, the average number of colleges per 100000 population," I do not understand how this indicator is related to the digital economy. This needs to be justified.

6. Conclusion

l.553-554: “unicorn enterprises and gazelle enterprises". Are these terms commonly used ? If not, I suggest inserting a short definition.

Reviewer 2 Report

After reading thoroughly of this paper, it is clear, to the point, and systematic written manuscript. The introduction, theoretical analysis ,research hypotheses, models, variable selection, and data sources are good.  There are some issues are associated with this paper and specific comments are:

1. Page No:1 and Line No. 33 – 34 give sources to “China has been the country with the highest carbon emissions in the world”, please provide reference.

2.Your Replace Econometric Model indicates that the digital economy has significantly reduced carbon emission, but the Instrumental Variable Method, the results shows that digital economy has a significant negative impact on carbon emissions. How it is correlated with the digital economy and carbon emissions?

3. Policy Implications: on page no. 16-18, Para 3, line 556:  the Chinese government announced many green development initiatives to reduce carbon emissions.   

Once these questions are addressed, the article should be published.

https://www.sciencedirect.com/science/article/pii/S232542622200047X
